# OpenReview forum: "Scaling depth capacity via zero/one-layer model expansion"
_ICML.cc/2026/Conference — ICML 2026 regular_

### Official Review · Reviewer_sVQm · 2026-02-22

**Soundness:** 3
**Presentation:** 1
**Significance:** 3
**Originality:** 3
**Overall Recommendation:** 4
**Confidence:** 3

**Summary:**

This paper proposes Deep Progressive Training for scaling depth while reducing training compute: train an extremely shallow “source” model (as extreme as 0-layer / 1-layer) for a portion of training, then perform a single depth expansion at time \tau to the full target depth and continue training. This also studied how/where/which/when should do depth expansion.

**Compliance With Llm Reviewing Policy:**

Affirmed.

**Final Justification:**

Considering the strengths and weaknesses, I would lean toward a weak accept.

**Key Questions For Authors:**

Questions

1. How exactly do you handle optimizer state at depth expansion (momentum / normalization stats / any optimizer-specific state) for both existing parameters and newly added layers? Is state reset/initialization important for mixing time?
2. How do you handle the batch_norm layer while expanding the depth of rennet?
3. How stable is the estimated mixing time across model sizes/architectures?
4. Will de-expansion/contraction will be more efficient? Let’s think about the case after expansion, the model is well-trained with 1B tokens, then you can fold/contraction it to small one but do not hurt its performance? like [1] which seems more efficient.

[1] Zou, Longwei, Han Zhang, and Yangdong Deng. "A multi-level framework for accelerating training transformer models." arXiv preprint arXiv:2404.07999 (2024).

**Limitations:**

No. More details mentioned in the weakness part should be discussed, like diverse random seeds, model scales, datasets, optmizers, real efficiency, and so on.

**Strengths And Weaknesses:**

Strengths

- The discussion of proposed method is clear and well. The paper meaningfully dissects the roles of initialization, schedule (WSD vs cosine), and expansion timing /tau mixing time, rather than presenting a single cherry-picked curve.
- Results span multiple LLM families and include ResNet/ImageNet; the paper also discusses/validates behavior on MoE.
- There are some theoretical grounding. the convex analysis provides a coherent lens linking empirical observations  to a conceptual mechanism.


Weaknesses

- Writing clarity is a major issue. The information density of paragraphs is too high. The narrative introduces symbols/terms (e.g., mixing-related quantities, 6BTN) before adequate motivation/definitions, making the paper harder to follow than necessary.
- It’s hard to follow figs. Many plots show multiple colored curves but captions/legends do not consistently map each curve to a precise setting.
- The key claims appear tightly coupled to the optimizer/setup and the WSD schedule. It remains unclear how robust the headline effects are under standard LLM training recipes.
- The paper does not clearly state whether key results are averaged over multiple random seeds, nor does it report variance/error bars. For example, in the multi-layer expansion methods comparison ( Fig. 4),  copying_last seems not far away from copying-all methods, which may weaken the claim “copying all layers is consistently better than only copying one layer”.
- The paper evaluates efficiency mainly via FLOPs-based proxies. In practice, training very small models can reduce GPU utilization and become overhead/communication-bound, so FLOPs savings may not translate to proportional wall-clock speedups.

---

### Official Review · Reviewer_Mw8G · 2026-03-09

**Soundness:** 2
**Presentation:** 3
**Significance:** 3
**Originality:** 3
**Overall Recommendation:** 4
**Confidence:** 2

**Summary:**

This paper studies a progressive training strategy in which a zero- or one-layer model is trained first and then expanded into a deeper model during training. As a concrete example, the authors show that, for GPT-2, this approach can reduce training compute by about 80% while achieving validation loss comparable to that of a fully trained deep model. Through extensive experiments, the paper investigates how performance depends on factors such as the learning-rate schedule and the initialization scheme for newly added layers, including random initialization and layer copying, and identifies effective design choices for depth expansion. The paper also provides a convergence analysis under convex, Lipschitz losses, and argues that progressive training can be viewed as “projected gradient descent (PSG) on a masked deep model + initialization of new layers + SGD.”

**Compliance With Llm Reviewing Policy:**

Affirmed.

**Final Justification:**

The authors did not provide a rebuttal; therefore, my original evaluation and score remain unchanged.

**Key Questions For Authors:**

* In Section 4, what justifies modeling the optimization dynamics as convex-like, especially immediately after depth expansion?
* Can the authors provide either theoretical or empirical evidence that the optimum (W^*) of the expanded model admits a decomposition of the form ($W^\* = [w^\*, x^\*]$)?
* If these issues cannot be satisfactorily addressed, and the analysis in Section 4 turns out to be less well justified than claimed, are there alternative theoretical perspectives that could still support the paper’s main empirical findings?

**Limitations:**

Yes.

**Strengths And Weaknesses:**

### Soundness

The effectiveness of the proposed depth expansion strategy is supported by a large number of experiments, and I believe the empirical soundness of the method is well validated. On the other hand, I have concerns about the analysis in Section 4. In this section, the authors argue that progressive training can be viewed as PGD, but this result seems to rely heavily on the assumption in line 302: “simplify the analysis by assuming (W^* = [w^*, x^*]).” However, there is no guarantee that this assumption actually holds. In general, the optimum (W^*) of the larger model does not necessarily decompose into the optimum (w^*) of the smaller model concatenated with some additional parameters (x^*). Since the subsequent analysis appears to depend strongly on whether this decomposition is valid, the authors should verify whether this assumption is actually satisfied, at least approximately, in trained models.

In addition, the authors state that the inequalities in Sections 4.3 and 4.4 are tight based on Bu et al., 2026. However, the soundness of this claim would be better supported if they could also verify this experimentally in the present paper.

Moreover, in Section 4.1, the authors motivate their analysis by stating that the training dynamics of deep learning resemble those of convex optimization. However, the cited work, Bu et al., 2026, appears to suggest that such convex-like behavior emerges after some amount of training has already taken place. In contrast, in this paper the loss becomes very large immediately after depth expansion, and it is unclear whether the optimization dynamics at that stage can still be reasonably regarded as convex-like.

### Presentation

The paper is well organized and generally easy to read.

### Significance

The proposed method achieves performance competitive with that of training the full target model from scratch, while requiring only a very small fraction of the original training cost. I find this to be a strong and significant result.

### Originality

I am not very familiar with the progressive learning literature, so I cannot confidently judge whether the level of originality is sufficient. However, based on my limited survey, the paper appears to contain novel contributions.

---

### Official Review · Reviewer_ZSqr · 2026-03-12

**Soundness:** 3
**Presentation:** 4
**Significance:** 3
**Originality:** 4
**Overall Recommendation:** 5
**Confidence:** 3

**Summary:**

This paper studies progressive depth expansion for large models and argues that an unexpectedly small source model—especially a zero-layer or one-layer model—can already provide a strong loss-compute tradeoff. The authors analyze how the initialization of new layers, the learning-rate schedule, and the expansion time affect training, and use both experiments and a simplified optimization analysis to motivate their design choices. Based on these findings, the paper recommends a recipe built around zero/one-layer single-stage expansion, WSD scheduling, and random/copy-based initialization. Experiments on language, vision, and some MoE models suggest that this strategy can substantially reduce training compute while maintaining similar final performance to fixed-size training.

**Compliance With Llm Reviewing Policy:**

Affirmed.

**Final Justification:**

The authors did not provide a rebuttal

**Key Questions For Authors:**

1. How robust are the main conclusions under AdamW-style training?
The paper’s final recipe is centered on Muon-NSGD and WSD. Since AdamW remains common in large-scale language-model training, it would be helpful to clarify whether the main findings—especially the effectiveness of zero/one-layer expansion and the mixing behavior—also hold under AdamW-style setups.

2.Is WSD necessary, or mainly the best choice among the schedules tested here?
The paper makes a convincing case that WSD improves the loss-compute tradeoff, but it is less clear whether the method fundamentally depends on WSD or whether WSD is simply the strongest option among the schedules evaluated.

**Limitations:**

The paper would benefit from a more explicit discussion of its limitations：
(i) the dependence of the proposed recipe on Muon-NSGD / muP-scaled optimization rather than more standard AdamW-style setups
(ii) the gap between the convex SGD-based theory and actual non-convex large-model pretraining

**Strengths And Weaknesses:**

The paper addresses an important problem, namely how to reduce the training cost of deep models without sacrificing much final quality, so the significance is clear. The main empirical finding—that zero/one-layer expansion can already work well—is interesting and gives the paper a meaningful degree of originality. In terms of presentation, the paper is generally well organized and easy to follow, especially in how it connects initialization, schedule choice, and expansion timing. For soundness, the paper combines empirical evidence with a theoretical perspective, and the overall story is coherent.

---

### Official Review · Reviewer_DGSW · 2026-03-17

**Soundness:** 3
**Presentation:** 2
**Significance:** 3
**Originality:** 3
**Overall Recommendation:** 4
**Confidence:** 3

**Summary:**

The authors study variants of progressive training, where a shallow transformer is trained first and then expanded by replicating or adding layers before continuing training. The authors find and evaluate a number of factors that can positively impact this method, specifically random or copy initialization, WSD schedule, single stage expansion, and using variants of muon. The final result shows an impressive 5x speedup.

**Compliance With Llm Reviewing Policy:**

Affirmed.

**Key Questions For Authors:**

N/A

**Limitations:**

There is limited discussion of limitations, and given the surprisingly strong results this would be important to highlight

**Strengths And Weaknesses:**

Strengths

— Progressive training appears to be a promising direction not well understood by the community to gain efficiency in pretraining

-- The results are impressive, 5x acceleration with significant experimental coverage

Weakness
--  Although there is a comprehensive variations of architectures studied the largest scale of training are not considered (e.g. more modern LLM at 7B for longer token budgets)

-- There is a particular recipe of scheduler and optimizer that seems to work well, this might limit the utility of the method

-- The theory is in a relatively simple setting. More importantly the theory seems to have a flaw.

-- The results seem very impressive and it would be important for the authors to highlight any limitations. Additionally code to reproduce the experiments would

---

### Decision · Program_Chairs · 2026-04-30

**Decision:**

Accept (regular)

**Comment:**

This paper studies progressive depth expansion for training large-scale models, demonstrating that scaling up from an unexpectedly small zero-layer or one-layer source model yields a strong tradeoff between computational cost and loss. The authors provide a comprehensive training recipe centered around random or copy initialization for new layers, the Warmup-Stable-Decay (WSD) learning rate schedule, and the Muon-NSGD optimizer. Through empirical evaluations across diverse architectures, it shows that this strategy can accelerate training by up to 5x (saving approximately 80% compute) while maintaining final performance comparable to fixed-size training. Additionally, the paper offers a theoretical perspective based on convex optimization to justify its design choices and explain the mixing behaviors of the loss.

The initial reviews were unanimously positive, with all four reviewers recommending acceptance (three "Weak Accept" and one "Accept" score). Reviewers consistently praised the significance and originality of the empirical results, highlighting the compute savings and the ablation of expansion strategies. However, several shared concerns emerged. Reviewers questioned the robustness and generalizability of the proposed recipe, specifically whether the strong results heavily depend on the WSD schedule and Muon optimizer, or if they hold under standard AdamW setups. Furthermore, multiple reviewers raised validity concerns regarding the theoretical analysis in Section 4: that the convex optimization assumptions (e.g., $W^* = [w^*, x^*]$) might not hold in practice, especially during the highly non-convex loss spikes immediately following depth expansion.

Unfortunately, the authors missed the official rebuttal deadline due to a misunderstanding of the platform's functionality and were locked out of the subsequent discussion period. Consequently, the reviewers' initial concerns remain unaddressed. This includes the questions regarding theoretical assumptions, the reliance on specific optimizers, the lack of variance reporting across random seeds, and the distinction between theoretical FLOPs savings and actual wall-clock speedups. Reviewers explicitly noted the absence of a rebuttal in their final justifications, and as a result, maintained their original scores without modification.

Despite the lack of an author rebuttal and the lingering questions surrounding the theoretical framing, the core empirical contribution of the paper remains compelling. Demonstrating a 5x acceleration in pretraining while maintaining competitive loss across a diverse set of modern architectures is a significant achievement that researchers are likely to build upon. The weaknesses identified by the reviewers mainly limit the breadth of the theoretical claims rather than invalidating the central empirical findings. Therefore, weighing the strong empirical results against the unaddressed theoretical and presentation concerns, the final recommendation is to Accept the submission. The authors are encouraged to incorporate the reviewers' feedback into the camera-ready version, specifically by softening the theoretical claims, explicitly discussing the limitations regarding optimizer reliance, and improving the clarity of the figures.